# Short Histone H2A Variants: Small in Stature but not in Function

**DOI:** 10.3390/cells9040867

**Published:** 2020-04-02

**Authors:** Xuanzhao Jiang, Tatiana A. Soboleva, David J. Tremethick

**Affiliations:** The John Curtin School of Medical Research, The Australian National University, P.O. Box 334, 2601 Canberra, Australia; xuanzhao.jiang@anu.edu.au

**Keywords:** chromatin, nucleosomes, histone variants, H2A.B, H2A.L, acidic patch, splicing, histone-protamine exchange, chromatoid bodies, piRNA

## Abstract

The dynamic packaging of DNA into chromatin regulates all aspects of genome function by altering the accessibility of DNA and by providing docking pads to proteins that copy, repair and express the genome. Different epigenetic-based mechanisms have been described that alter the way DNA is organised into chromatin, but one fundamental mechanism alters the biochemical composition of a nucleosome by substituting one or more of the core histones with their variant forms. Of the core histones, the largest number of histone variants belong to the H2A class. The most divergent class is the designated “short H2A variants” (H2A.B, H2A.L, H2A.P and H2A.Q), so termed because they lack a H2A C-terminal tail. These histone variants appeared late in evolution in eutherian mammals and are lineage-specific, being expressed in the testis (and, in the case of H2A.B, also in the brain). To date, most information about the function of these peculiar histone variants has come from studies on the H2A.B and H2A.L family in mice. In this review, we describe their unique protein characteristics, their impact on chromatin structure, and their known functions plus other possible, even non-chromatin, roles in an attempt to understand why these peculiar histone variants evolved in the first place.

## 1. Introduction

In eukaryotes, DNA is assembled into a highly complex and dynamic structure called chromatin. Chromatin is constructed from the universal repeating protein–DNA complex, the nucleosome core [1]. Nucleosome cores are comprised of two tight super helical turns of DNA (146 bp of DNA) wrapped around a disk-shaped protein assembly of eight canonical histone molecules (two molecules each of the core histones H2A, H2B, H3, and H4). Histones are small proteins that are highly basic rich in lysine and arginine residues and, through electrostatic interactions, neutralise the negatively charged DNA, thus enabling the DNA to be tightly compacted [1].

In higher eukaryotes, each major canonical histone is expressed from multiple copy genes that are highly similar and are among the slowest evolving proteins [2]. This highlights that their structure and function must be retained for cell survival, i.e. to maintain the core structure of chromatin and chromosomes. Core histones are highly expressed in S phase, which is expected given that chromatin needs to be reassembled following DNA replication [3]. 

On the other hand, nonallelic variants exist for all canonical histones that have major differences in their primary sequence, and have evolved at different rates [4,5,6]. For example, the ubiquitous histone H2A variant H2A.Z is evolutionarily conserved (from budding yeast to humans), while the short H2A histone variant class appeared later in evolution, in eutherian mammals, and is rapidly evolving [4,6]. This implies that H2A.Z has an important conserved function, e.g., in mitosis [7,8,9,10], whereas the short H2A variants may have highly specialised and/or unique chromatin functions required for evolutionary advanced mammals.

In addition, unlike canonical histones, histone variants are expressed from a small number of genes (usually two or three that lack introns) [4,5,6] and are expressed throughout the cell cycle and in non-dividing cells [11]. This enables chromatin to be remodelled at any stage of the cell cycle or during certain phases in development. Short histone H2A variants can dramatically alter the structural properties of chromatin as well as having specific protein interacting partners (see below). 

The largest number of histone variants belong to the H2A class. This includes H2A.Z, H2A.X, MacroH2A, H2A.J and H2A.R, as well as the short histone variants H2A.B, H2A.P, H2A.L and H2A.Q [5,12,13]. The short H2A variants are expressed in the testis (and, in the case of H2A.B, also in the brain), thus they presumably have important roles during spermatogenesis (see below) [5,14]. An intriguing question is: Why do most of the core histone variants belong to the H2A class? The most likely answer is that H2A plays an important role in the nucleosome and in the assembly of chromatin. H2A contains a C-terminus docking domain (Figure 1) that stabilises the histone octamer by tethering the H3-H4 tetramer with the H2A–H2B dimer. The two H2A molecules (the L1 loop regions) in a nucleosome interact with each other to stabilise the two turns of the DNA helix at the back face of the nucleosome [1,13,15,16,17]. Importantly, H2A also contributes significantly to the nucleosome surface, thereby having a major role in mediating the intra-nucleosome-nucleosome interactions required for chromatin compaction [15,18], as well as being a docking site for many chromatin factors and enzymes [19]. Therefore, H2A variants have the ability to regulate many aspects of the chromatin structure and function relationship.

The short H2A histone variant family are the most divergent of the histone H2A variant family (<50% amino acid identity with canonical H2A; Figure 1) [5,13]. The best characterised of these variants are mice H2A.B.3 (involved in gene activation and splicing) [14] and H2A.L.2 (required for the histone–protamine exchange process) [20] (see below). H2A.B was first identified by Chadwick and Willard in an EST human database and overexpression of this variant in female somatic cells revealed that it was excluded from the inactive X chromosome (the Barr body) [21]. Therefore, H2A.B was originally designated H2A.Bbd (Barr body-deficient). However, subsequent localisation experiments in its proper physiological context (in the mouse testis) revealed that H2A.B is in fact present on the inactive X chromosome following meiotic sex chromosome inactivation (see below) [22]. Therefore, histone variant overexpression studies in an irrelevant cell type should be interpreted cautiously.

H2A.L.2 were identified by Khochbin and colleagues using a proteomic approach in mice testes [23]. An important distinguishing feature of different short histone H2A variants expressed in the same species is their timing of expression during spermatogenesis. For example, in the mouse, H2A.B.3 is maximally expressed following the completion of meiosis (in round spermatids) [22], while H2A.L.2 is maximally expressed at later stages (condensing spermatids) when histones are bound by transition proteins and then replaced with protamines [20,23].

## 2. Short Histone H2A Variant Evolution 

Human H2A.B is encoded by three intronless non-allelic genes (*H2AFB1*, *H2AFB2*, and *H2AFB3)* and are all localised in the q28 locus of the human X chromosome [5,24]. The murine homolog H2A.B.3 (previously H2A.Lap1 for lack of an acidic patch) is also encoded by three X chromosome genes (*Gm14920*, *H2afb2*, and *H2afb3*) but are located at different syntenic locations compared to the human X chromosome [5,22]. Evolutionary, the four classes of short H2A variants *H2A.B*, *H2A.L*, *H2A.P* and *H2A.Q* genes appear to have resided (and still remain) on the X chromosome at least since the common ancestor of all eutherian mammals [5,22]. However, rodents appear to be the exception for this X chromosome location. One mouse *H2a.l* gene relocated to the Y chromosome while a second moved to chromosome 2 [5,22]. Interestingly, ancestral genes encoding H2A.B moved away from the X chromosome to autosomes, but these genes decayed while the rat–mouse common ancestor acquired a new functional *H2A.B* gene in a new X chromosome location [5,22]. This suggests that their X-linkage may be evolutionarily preferred. In fact, genes expressed during spermatogenesis are over-represented on the X chromosome [25].

Interestingly, an ancestor histone H2A variant branched out to H2A.R (still possessing a C-terminus) in monotremes and marsupials, and to the four classes of the short histone H2A variants [5]. In addition, intriguingly, it has been proposed that the common ancestor of eutherian mammals encoded two or three H2A.B genes, three H2A.L genes, a single H2A.P gene, and a single H2A.Q gene in six distinct X-linked loci [5]. However, no current mammalian genome contains the same repertoire of functional short histone H2A variants since each mammalian species has lost one or more of the four classes of short H2A variants. For example, humans have retained H2A.B and H2A.P but lost H2A.Q and H2A.L, while mice lost H2A.Q.

These observations suggest that no individual short H2A variant performs a universal and essential function in eutherian mammals, or that different short H2A variants can perform similar functions in different mammals. For example, H2A.L.2 is essential for mouse sperm viability [20] and therefore H2A.B or H2A.P may perform this function in humans. Alternatively, it is formally possible that the function that H2A.L.2 plays in mice is not required in humans and is replaced by a different “non-histone variant” mechanism (see Section Concluding Remarks). To our knowledge, no studies have examined the evolution (conservation versus diversification) of the promoter DNA sequences of short histone H2A variant genes, which could help explain their testis-restricted and stage-specific gene expression patterns.

## 3. Features of Short Histone H2A Variant Proteins and Their Impact on Chromatin Structure

The protein features that define short H2A variants compared to canonical H2A have a profound effect on nucleosome and higher-order chromatin structures. As previously noted, these histone variants lack a C-terminus and the last segment of the docking domain of H2A (Figure 1A). The C terminus of H2A contacts the N-terminal α helix of histone H3 in the nucleosome core. This α helix and its preceding unstructured N-terminal tail is responsible for binding and guiding the DNA that enters and leaves the nucleosome core [1]. Not surprisingly, assembly of mouse H2A.B.3 and human H2A.B into nucleosome cores results in the unwrapping of the DNA ends (~10–20 bp) from the nucleosome surface producing a highly unstable structure [18,26,27,28,29,30,31,32,33]. As a consequence of this, only ~120 bp of DNA is protected against digestion by micrococcal nuclease compared to 146 bp in canonical nucleosomes [18,27]. This insufficient DNA wrapping leads to a distinctive H2A.B nucleosome structure where the two DNA nucleosomal ends are extending away from each other instead of forming the classic V-shape [28]. Similarly, a H2A.L.2 nucleosome only wraps ~130 bp DNA [20,34], suggesting that all short histone variants would alter nucleosome structure in this way.

Notably, this unwrapping of DNA from the nucleosome surface may also prevent linker histone H1 binding [30]. Linker histones bind to the DNA ends of a nucleosome and the adjoining linker DNA to promote chromatin compaction. Therefore, this may provide an additional mechanism to ensure that short H2A histone variant-containing nucleosomes and chromatin would always remain open and accessible. These in vitro characteristics are consistent with fluorescence recovery after photobleaching (FRAP) experiments, where green fluorescence protein (GFP) tagged-H2A.B exchanged significantly more rapidly than GFP-H2A when overexpressed in HeLa cells [35,36]. Biochemical fractionation experiments revealed that approximately half of GFP-H2A.B was chromatin bound with the remainder found in the soluble non-chromatin bound fraction in the nucleus [35,36].

Significant amino acid residue differences also occur in the L1 loop region, suggesting that the formation of stable hybrid nucleosomes with H2A is unlikely [5,13,16]. This is a common feature of different histone H2A variants. For example, H2A.Z also diverges significantly in the L1 loop region compared to H2A and heterotypic H2A.Z-H2A nucleosomes are unstable [16,37]. Despite this structural knowledge, the full impact on nucleosome structure will require the determination of the high-resolution X-ray structure of a short H2A histone variant-containing nucleosome core particle (which will be challenging given the instability of a H2A.B-containing nucleosomes).

However, recently, the H2A.B-H2B dimer structure (created as a single fusion protein) was solved to 2.6-Å resolution [38]. The most significant difference compared to H2A was a transition in the long C-terminal α helix of H2A to a less stable short 3_10_ helix thus revealing a major change in the structure of the remaining docking domain [38]. This could impact not only the stability of the histone octamer but also destabilise intra-nucleosome–nucleosome interactions because this C-terminal α helix juxtaposes the acidic patch [1].

The acidic patch (comprised of six acidic amino acids of H2A and two acidic amino acids of H2B located on the nucleosome surface) drives the compaction of nucleosomal arrays into chromatin higher-order structures by promoting intra-nucleosome–nucleosome interactions. This is driven primarily by an interaction between the acidic patch and a histone H4 N-tail originating from a neighbouring nucleosome [15,39,40]. H2A.B and H2A.B.3 (and H2A.L.2) lack this acidic patch (Figure 1A). Therefore, physiologically spaced nucleosome arrays containing H2A.B or H2A.B.3 cannot condense and hence are transcribed by RNA Pol II almost as efficiently as naked DNA in vitro [18,41]. Indeed, the extent of compaction of a H2A.B-containing array is no greater than a H3–H4 tetramer subnucleosomal array, i.e. an array assembled without H2A–H2B dimers [18]. Further, neutralising the acidic patch of H2A mimics these properties of H2A.B [18].

On the other hand, H2A.Z has an extended acidic patch, thus promoting chromatin compaction and transcriptional repression in vitro [18,40]. Therefore, regulation of the acidic patch by histone variants appears to be an important mechanism by which chromatin structure is modulated. The acidic patch of a canonical nucleosome is also a “hot spot” for the binding of a diverse range of chromatin interacting proteins and therefore the short histone H2A variant family would also be expected to modulate the binding of such nucleosome binding proteins [19].

The N-termini of the short histone variants also differ significantly from that of H2A (Figure 1B). Indeed, perhaps the most striking functional feature of H2A.B and H2A.B.3 is that its N-terminus is an RNA binding module. This provides a direct link between chromatin structure and RNA function (see below) [14]. The unstructured N-terminal tails of all the H2A.B family members lack lysine residues but are enriched with arginine, serine and glycine residues, which is a common feature of many RNA binding proteins that contain unstructured regions responsible for their RNA binding activity (Figure 1B) [42].

Comparison of short histone H2A variants expressed in the mouse (H2A.B, H2A.L and H2A.P) (Figure 1C) reveals that H2A.L.2 is also enriched with arginine residues while H2A.P is not, suggesting that the former but not the latter may also bind RNA. Indeed, Khochbin and colleagues recently revealed that H2A.L.2 is a RNA binding protein [43]. Further, they postulated that the major increase in RNA synthesis from major satellite DNA repeats at pericentric heterochromatin in condensing spermatids may contribute to the stabilisation of the interaction between H2A.L.2 and this compacted domain. However, while the RNA binding activity may be necessary for this interaction, it appears to be not sufficient since their study also revealed that other aspects of H2A.L.2 are required for its targeting to pericentric heterochromatin [43].

While the N-terminal tails of H2A.B and H2A.L are similar, they differ significantly from H2A.P (Figure 1C) and H2A.Q [5], suggesting that short histone H2A variants do not share the same function given the importance of the N-terminal tails in regulating nucleosome function. Further, there are also many other significant amino acid residue differences between the different short histone variant classes in their histone fold domains [5]. Therefore, while the overall role of short histone H2A variants may be to destabilise chromatin, this property may be utilised in different ways to regulate chromatin function. To date, no post-translational modifications have been reported for these variants, which would add another layer of control.

## 4. The Role of H2A.L.2 in the Exchange of Histones with Protamines

Spermatogenesis is a remarkably sequential process that converts mitotically dividing spermatogonia stem cells into differentiated haploid spermatozoa. A subpopulation of rapidly dividing stem cell spermatogonia stop dividing and differentiate into primary spermatocytes, which enter the first meiotic prophase. Cells then complete division I of meiosis to produce secondary spermatocytes, which then undergo meiotic division II to produce highly transcriptionally active haploid round spermatids. Differentiation continues, involving the shut-down of transcription and the exchange of histones with protamines, to ultimately produce mature spermatozoa. Not surprisingly, this process involves dramatic nuclear and chromatin remodelling events and, therefore, there are many opportunities for short histone H2A variants to function [44].

As mentioned above, one key aspect of the function of mouse short histone variants is the timing of their expression in the testis. H2A.B.3 expression begins during meiosis (the late pachytene stage) and peaks in post-meiotic round spermatids [14,22]. When round spermatids begin to elongate, H2A.B.3 is exported out of the nucleus, which is concurrent with the cessation of transcription [14]. At this stage, H2A.L.2 begins to accumulate with transition proteins. H2A.L.2 expression peaks at the subsequent condensed spermatid stage when protamine incorporation occurs [20,23]. Interestingly, H2A.L.2 remains in mature spermatozoa but disappear quickly from paternal chromatin following sperm-egg fusion, perhaps facilitating the remodelling of paternal chromatin in the zygote [45].

H2A.L.2 knock out mice are infertile because transition proteins can no longer associate with chromatin [20]. Specifically, the ability of H2A.L.2 to destabilise and open the nucleosomal DNA ends enables chromatin invasion by transition proteins. This transition protein–H2A.L.2-containing disrupted chromatin structure then provides the proper interface for protamine binding and subsequent histone displacement (interestingly, histones remain bound at pericentric heterochromatin). Significantly, H2A.L.2 preferentially dimerises with the H2B testis-specific variant TH2B and it is this double histone variant-containing nucleosome that is the preferred substrate for transition protein binding [20]. H2A.L.2 may also have additional roles as it has been suggested that it may function in regulating the apoptosis of spermatogenic cells, which is critical for precise homeostasis of the different cell types in the testis [46].

Most interestingly, the incorporation of H2A.L.2 into chromatin is impaired in H2A.B knock out mice and, not surprisingly, the incorporation of transition proteins is also dysregulated, even though H2A.B and H2A.L.2 are not expressed at the same time [47]. Therefore, the function of one histone variant (H2A.L.2) is dependent upon another histone variant (H2A.B.3) in a yet to be described way.

## 5. Role of H2A.B in Gene Activation and Splicing

In addition to H2A.L.2, the only other short histone H2A variant that has been studied in its proper physiological context is H2A.B. As noted above, during spermatogenesis, the highest overall level of gene expression occurs at the round spermatid stage [48] (where much of the produced mRNA is stored but translated at later stages when transcription ceases), which correlates with the time of maximally H2A.B.3 expression [16,22]. This suggested a role for this variant in the activation of gene expression. Indeed, a combination of ChIP-Seq and RNA-Seq experiments revealed that H2A.B.3 is targeted to the transcription start site (TSS) of genes, concurrent with their transcriptional activation [14,16,22]. Notably, the incorporation of H2A.B.3 at the TSS of genes already active prior to the round spermatid stage correlated with even higher levels of gene expression [16,22,49]. The location of a H2A.B.3 nucleosome at the TSS was initially unexpected because the view was that an active TSS was nucleosome- free [16]. However, it is now known that this is not the case [50]. The positive correlation between H2A.B.3 incorporation at the TSS and gene expression was also observed in the mouse hippocampus [14].

The X chromosome becomes inactivated during the pachytene stage and remains in an inactive state throughout spermatogenesis [51]. However, ~13% of X-linked genes reactivate in the round spermatid stage [51] and, importantly, H2A.B.3 is targeted to their TSSs concomitant with their transcriptional activation [22]. This leads us to suggest that H2A.B.3 is able to co-ordinately activate the expression of many genes in round spermatids by being targeted to and directly opening the chromatin region encompassing their TSS. Further supporting a role in promoting transcriptional initiation, H2A.B.3 ChIP-mass spectrometry experiments revealed an interaction with the initiation form of RNA Polymerase II [14]. One caveat to this hypothesis is that no specific chaperone capable of directing H2A.B.3 to the TSS has been identified to date.

H2A.B is not only located at the TSS of active genes but also throughout the gene body including at intron–exon (and exon-intron) boundaries [14]. This suggests that H2A.B may also play a role in the elongation of transcription (consistent with its ability to decompact chromatin) and/or pre-mRNA splicing in round spermatids. 

There is significant evidence to support a role for H2A.B.3 in splicing in mouse round spermatids: (1) H2A.B.3 ChIP-mass spectrometry experiments identified an interaction between chromatin-bound H2A.B.3 and a range of spliceosome subunits and splicing factors. Intriguingly, binding of H2A.B.3 with splicing factors is in competition with its ability to bind RNA (see model depicted in Figure 2) [14]. (2) H2A.B.3 is enriched in splicing speckles. In round spermatids, splicing speckles are not only a storage and assembly site for pre-mRNA splicing factors, but also appears to be the site for active transcription [14]. (3) H2A.B.3 at intron–exon boundaries directly interacts with transcript RNA [14]. (4) H2A.B knock out mice display a major change in splicing events [47]. Further, correlated with these splicing changes, there was a loss of proper RNA Pol II targeting to splicing-transcription speckles [47]. Finally, it was also previously shown that the exogenous expression of H2A.B in HeLa cells can both interact with splicing factors and alter pre-mRNA splicing confirming that H2A.B has the intrinsic ability to modulate splicing outcomes [31].

The histone variant H2A.Z is also found at intron–exon boundaries but on inactive genes. Therefore, during transcription, there is a histone variant replacement process whereby H2A.B.3 replaces H2A.Z [14]. Based on these findings, we propose the following model (Figure 2). Following the replacement of H2A.Z with H2A.B.3 to assemble active chromatin, H2A.B.3 directly recruits splicing factors from splicing speckles to intron–exon boundaries. Upon transcriptional elongation and the synthesis of mRNA, H2A.B.3 binds and “holds” onto the RNA thus releasing the splicing factors to facilitate the splicing process. Taken together, the data suggest that H2A.B is a unique epigenetic regulator that can bind to DNA, RNA and proteins to facilitate different steps in the gene expression pathway. 

## 6. Other Roles of H2A.B

Our previous analyses of H2A.B.3 interacting proteins in round spermatids not only revealed an interaction with RNA processing factors, but also, intriguingly, with proteins involved in the piRNA pathway [14]. Supporting this previous observation, we now show that immunoprecipitation of H2A.B.3 also brings down mouse Piwi protein (Miwi) (Figure 3A). This has led us to reanalyse H2A.B.3 localisation in mouse haploid round spermatids. Most interestingly, while the bulk of H2A.B.3 is chromatin-bound, there is a small non-chromatin bound fraction of H2A.B.3 present in a large ribonucleoprotein granule known as a chromatid body (CB) (Figure 3B). This CB contains Miwi and Dicer (Figure 3B) that previously lead to the view that this special granule functions as a germ cell-specific RNA processing centre [48,52,53]. An interesting feature of CBs is its active and non-random movements around the cytoplasm of round spermatids making frequent contacts with nuclear pores enabling proteins and RNA to traffic between CBs and the nucleus [48,52,53]. Based on these observations, it is attractive to suggest that the unique RNA- and/or protein-binding ability of H2A.B.3 may enable it to have a non-chromatin role, perhaps as a chaperone, in the regulation of RNA function. The availability of H2A.B knock out mice should allow this prediction to be tested.

Ectopic H2A.B expression studies have showed that tagged H2A.B rapidly and transiently deposits at sites of DNA replication and DNA repair [26,54]. Supporting the former finding, a rigorous mass spectrometry analysis revealed an association between H2A.B and DNA replication factors [54]. A common feature of these sites of incorporation is that DNA is transiently exposed, providing a “window of opportunity” for H2A.B to be incorporated into nascent chromatin as it is being assembled (while canonical histones and other histone variants have specific chaperones for their controlled assembly into chromatin in somatic cells, H2A.B would not). Possibly, such an opportunistic mechanism of H2A.B incorporation can account for its targeting to active genes in round spermatids. This seems unlikely, however, given that, in round spermatids, H2A.B is targeted to specific regions on an active gene, i.e., the TSS, and H2A.B is not incorporated into all active genes [14,22].

As discussed above, a caveat in analysing the outcomes of H2A.B over-expression studies is that, in addition to non-physiological somatic cell types being used in these studies, the level of exogenous expression of this variant often far exceeds the relative level observed in round spermatids. Perhaps even more concerning, one study revealed that the over expression of H2A.B in HeLa cells (a commonly used cell line) caused DNA-replication dependent and independent damage and NF-κB-dependent apoptosis, which could impact any conclusion made regarding how H2A.B directly regulates gene expression in cell lines [55].

It is well established that drastic alterations to the epigenome occur in virtually all types cancers, and that various somatic cancers display the aberrant activation of germ cell-specific genes (known as cancer testis (C/T) antigens) [56,57]. This raises the question of whether short histone H2A variants may have a role in oncogenesis. While this remains an open question, H2A.B has been shown to be abnormally expressed in Hodgkin lymphoma (HL), a common blood cancer that affects adults of all ages [54,58]. The role of H2A.B in HL is currently unknown but one prediction might be that it regulates the expression of genes that are required to promote oncogenesis and cancer survival. It is known that, in different HL cell lines, the level of H2A.B expression directly correlates with the rate of proliferation [54]. Several studies have shown that, when H2A.B is ectopically expressed, it accumulates in the nucleolus [59,60]. Increased ribosomal biogenesis correlates with high cell proliferation rates and more aggressive cancers [61]. Alternatively, over-expressed H2A.B may simply accumulate in the nucleolus to be ultimately degraded. Future studies are required to examine a possible role for H2A.B in cancer.

## 7. Concluding Remarks

The discovery of short histone H2A variants has clearly added to the diverse repertoire of important functions of histone variants. Nevertheless, key questions remain: What was the driving force for their evolution? Why are they only expressed in the testis? Do the different combination of family members expressed in different eutherian mammals perform the same or different functions?

Clearly, H2A.L.2 is expressed in the mouse testis because it is essential for the histone to protamine exchange reaction but does H2A.B or H2A.P perform the same function in humans or are short histone H2A variants not required for this exchange reaction? Perhaps human and mouse spermatogenesis are different in subtle ways, as indicated by the fact that nucleosome retention in mature sperm differs markedly between them (~1% in mice and ~15% in humans) [44].

On the other hand, H2A.B is expressed at the same stage of spermatogenesis in mice and humans (unpublished data; manuscript in preparation), but why is it required to regulate transcription and splicing in the testis, and brain, but in no other cell type? Perhaps the answer lies in the fact that the testis, and brain, are two tissues known to be highly dependent upon RNA splicing to perform their complex functions and H2A.B may enhance the fidelity of these processes [62]. Alternatively, perhaps the key role of H2A.B is a non-chromatin-related function involving RNA that is unique in the testis and the brain, e.g., piRNAs. H2A.B knock out mice are not fully infertile [47] but perhaps in nature, the evolution of short histone H2A variants enabled the process of spermatogenesis to adapt to environmental challenges and other types of stresses. 

Sex-biased inheritance patterns of sex chromosomes can create a conflict over which sex is preferred, which in turn can drive the evolution of male- or female-biased sex ratios [25,63]. Given the rapid evolution of short histone H2A variants, their location on the X chromosome and being expressed in the testis, it has been postulated that they may function to help prevent this conflict to ensure an even sex ratio [5]. For example, the assembly of spermatogenesis-required genes on the X chromosome into a special H2A.B-containing chromatin structure may protect the X chromosome from detrimental Y chromosome-linked factors. Opposing these putative Y chromosome influences, perhaps the rapid evolution of H2A.B is in turn helping the X chromosome to gain an advantage over the Y chromosome. In conclusion, while the importance of these short histone H2A variants has yet to be fully realised, these remarkable histone variants will continue to provide a new level of understanding of the complex relationship between chromatin structure and function.

## Figures and Tables

**Figure 1 cells-09-00867-f001:**
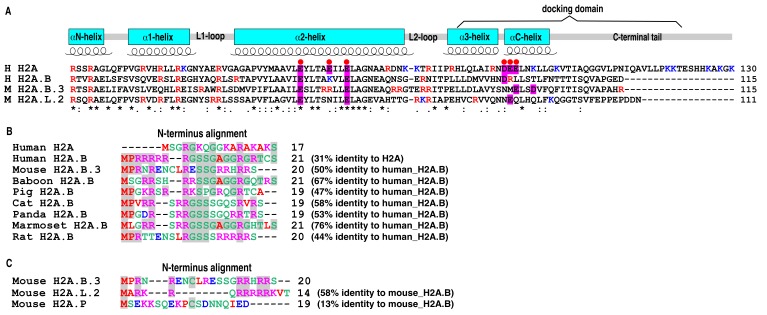
(**A**) Multiple protein sequence alignments of human H2A and H2A.B, and mouse H2A.B.3 and H2A.L.2. The structural features of the histone fold are annotated above the alignments. Arginine and lysine residues are highlighted in red and blue, respectively. The residues composing the “acidic patch” are shaded in magenta, and the “acidic patch” residues of H2A are annotated with red dot on top. The similarity of amino acid residues is indicated by the following symbols below the alignment: “*” (identical residues), “:” (residues with strongly similar properties) and “.” (residues with weakly similar properties). (**B**) Multiple amino acid residue alignments of the H2A.B N-termini of different eutherian mammals. Residues identical to those of human H2A.B are shaded in grey. (**C**) Alignments of the N-termini of all the short histone H2A variants expressed in the mouse. The residues identical to those of H2A.B.3 are shaded in grey. The amino acid residues are coloured following the default colour scheme of Clustal Omega (acidic residue, blue; basic residue, magenta; hydrophobic residue, red; polar residue, green). All multiple sequence alignments are generated using the EBI web server Clustal Omega (version 1.2.4) (EMBL-EBI, Hinxton, Cambridge, U.K) with default parameters. The sequence identity is calculated by Clustal Omega.

**Figure 2 cells-09-00867-f002:**
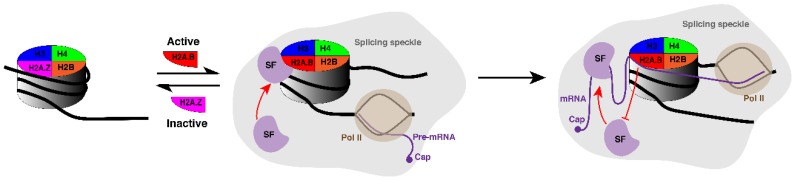
Model depicting the role of H2A.B in pre-mRNA splicing. Following the replacement of H2A.Z with H2A.B.3 to assemble active chromatin, H2A.B.3 directly recruits RNA processing factors from splicing speckles to an active gene. Upon transcriptional elongation and the synthesis of mRNA, H2A.B.3 binds and “holds” onto this RNA, thus releasing the splicing factors to facilitate the splicing process.

**Figure 3 cells-09-00867-f003:**
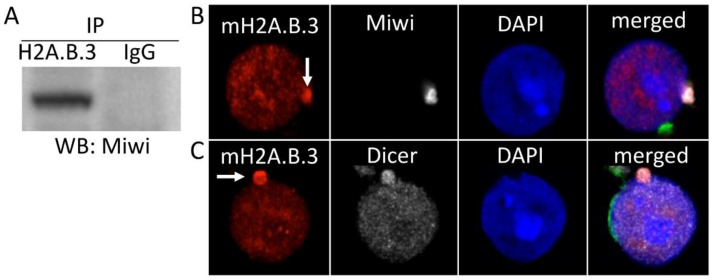
H2A.B.3 co-localises with Miwi and Dicer in chromatoid bodies of rounds spermatids and co-immunoprecipitates with Miwi protein. (**A**) H2A.B.3 co-immunoprecipitates with Miwi (Cell Signalling cat# 2079S) in non-crosslinked total testis lysates following the procedure described previously [14]. Non-targeting IgG control was used as a negative control during immunoprecipitation. (**B**,**C**) Hypotonic spreads of male germ cells [14] showing round spermatids indirectly immunostained with antibodies against mouse H2A.B.3 (red), lectin PNA (green), Miwi (**B**) (white) or Dicer (**C**) (white) (abcam cat# 13502) and counterstained with DAPI (blue) to visualize DNA. White arrows show the location of chromatoid bodies.

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
