# Peer review of "Short Histone H2A Variants: Small in Stature but not in Function"

_cells, 2020, doi:10.3390/cells9040867_

Round 1

Reviewer 1 Report

It was interesting for me to read this review. Hope, other journal readers will find this review useful also. 

At the same time, the manuscript requires revision. 

  1. Some sentences require correction (from my point of view), namely: lines 31-33 As the main ...; lines 38-40 Another feature ... ; lines 51-52 As will be...; lines 152-153 "the solving of a short..."; line 169-170 Further, ...; lines 254-255 "...currently, no specific...";  lines 269-270 "the ectopic expression..."; lines 297-299 Our previous....; lines 319-320 ...(this may occur...  
  2. Some terms should be verified: line 65  "non-histone binding proteins"; lines 146-147 "nuclear soluble compartment"; line 169 "tetramer (tetrasome?) array" ;  line 176 "binding proteins"; line 280 "those... to those...."; lines 279-280 "the N-termini of different eutherian 279 mammals" (please, clarify: identity of N-termini or identity of a whole sequence); line 310 "KO mice".
  3. Misprints should be corrected in the manuscript. 

Author Response

Thank you for your constructive comments that have improved the manuscript.

  1. Some sentences require correction (from my point of view), namely: lines 31-33 As the main ...;

lines 38-40 Another feature ... ; “Another feature has been removed”.

lines 51-52 As will be...; “As will be” has been removed and replaced with “see below”.

lines 152-153 "the solving of a short..."; this has been replaced with “Despite this structural knowledge, the full impact on nucleosome structure will require the determination of the high-resolution X-ray structure of a short H2A histone variant-containing nucleosome core particle”.

line 169-170 Further, ...;  “Further has been removed”

lines 254-255 "...currently, no specific..."; This has been removed and replaced with “One caveat to this hypothesis is that no specific chaperone capable of directing H2A.B.3 to the TSS has been identified to date.”

lines 269-270 "the ectopic expression..."; this has been replaced with “exogenous

 expression”

lines 297-299 Our previous....; This has been replaced with “Our previous analyses of H2A.B.3 interacting proteins in round spermatids not only revealed an interaction with RNA processing factors”

lines 319-320 ...(this may occur...; “This may occur” has been removed.

  1. Some terms should be verified:

line 65  "non-histone binding proteins"; This has been replaced with “many chromatin

factors and enzymes”.

lines 146-147 "nuclear soluble compartment"; This has been clarified by the following

“many chromatin factors and enzymes”.

line 169 "tetramer (tetrasome?) array" ; It is a tetramer array. Clarified by the following

“Indeed, the extent of compaction of a H2A.B-containing array is no greater than a

H3-H4 tetramer subnucleosomal array i.e. an array assembled without H2A-H2B dimers [18].”

line 176 "binding proteins"; Changed to “nucleosome binding proteins”.

line 280 "those... to those...."; “those” has been removed.

lines 279-280 "the N-termini of different eutherian 279 mammals" (please, clarify: identity of N-termini or identity of a whole sequence); Clarified -it is identity of N-termini.

line 310 "KO mice". Changed to “knock out”.

  1. Misprints should be corrected in the manuscript. We have performed several more editing checks and have found these. Thank you.

Reviewer 2 Report

The title of this review is The short histone variant H2A.B: small in stature but 2 not in function, but the manuscript talks about many different H2A variants. Overall this review contains some interesting information about histone variants of H2A, but the organization of the discussion is confusing to read. This manuscript needs a significant re-write.

This sentence on line 54 should be used in the abstract. "The largest number of histone variants belong to the H2A class"

Typo line 57 thet

The end of the introduction starting on line 77, seems to go off topic from an introductory paragraph.  I would recommend to include this information elsewhere in the manuscript. The introduction should end with a summary statement describing the purpose of this review. Then start sections for each histone short variant.

The figure legend for Figure 3 does not describe the figure, and needs more detail.

In the discussion for ‘Short histone H2A variant evolution’, it would be helpful if the authors included a figure of the evolutionary tree.

Line 54 states “The largest number of histone variants belong to the H2A class. This includes H2A.Z, H2A.X, 54 MacroH2A, H2A.J, H2A.R and the short histone variants H2A.B, H2A.P, H2A.L, H2A.Q”. Maybe a better way to organize this review would be to focus on what is known about the function of each variant specifically. 

It is unclear why the authors are focusing on H2A.B variants. Or at least that’s the title, but there is a whole section on The role of H2A.L.2 in the exchange of histones with protamines

My recommendation is to reject this manuscript.  They could re-submit after a major re-organization of the material.

Author Response

I have fixed the title, the typos, and the legend to the figure. I disagree with the other comments.

Reviewer 3 Report

This is an excellent, important and timely review on a specific group of H2A histone variants known as short variants. In fact, besides the original research articles reporting the functional studies of several members of this family, no review has specifically focussed on these data to propose a critical overview of their functions.

This review has therefore the merit to cover this area and should be of high interest to histone variant specialists and to chromatin biologists in general. It is well written with most of the arguments and discussions relevant and well justified. The literature cited is also complete and up to date.

However, the authors need to pay a particular attention to following points in the text.

1 – Line 36, …”their the”…Please delete “the”.

2- Line 36, it would be appropriate to cite reference 9 here along with references 4 and 5.

3- Line 47, what do the authors mean by “highly developed animals”?

4- Line 57, please replace “thet” by “they”.

5- Line 64, please replace “theintra-” by “the intra-”.

6- Lines 157, 160 and 330, “alpha” is missing.

Author Response

Thank you for picking up these mistakes. I have changed highly developed to evolutionary advanced.

1 – Line 36, …”their the”…Please delete “the”.

This has been deleted.

2- Line 36, it would be appropriate to cite reference 9 here along with references 4 and 5.

This has been done. Reference 9 now becomes reference 5.

3- Line 47, what do the authors mean by “highly developed animals”?

This has been changed to “evolutionary advanced mammals”.

4- Line 57, please replace “thet” by “they”.

Corrected

5- Line 64, please replace “theintra-” by “the intra-”.

Corrected.

6- Lines 157, 160 and 330, “alpha” is missing.

a helix is present.